# Determination of the Total Phenolic Content in Wine Samples Using Potentiometric Method Based on Permanganate Ion as an Indicator

**DOI:** 10.3390/molecules24183279

**Published:** 2019-09-09

**Authors:** Ziqi Sun, Yufeng Zhang, Xinyue Xu, Minglin Wang, Lijuan Kou

**Affiliations:** 1School of Pharmacy, Binzhou Medical University, Yantai 264003, China (Z.S.) (Y.Z.) (X.X.); 2School of Food Science and Engineering, Shandong Agricultural University, Taian 271018, China

**Keywords:** potentiometric, total phenolic content, wine, indicator

## Abstract

A rapid and accurate determination method for total phenolic content is of great importance for controlling the quality of wine samples. A promising potentiometric detection approach, based on permanganate ion fluxes across ion-selective electrode membranes, is fabricated for measuring the total phenolic content of wine. The results show that the presence of phenols, such as gallic acid, leads to a potential increase for the potentiometric sensor. Additionally, the present sensor exhibits a linear potential response with the concentration range from 0.05 to 3.0 g/L with a detection limit of 6.6 mg/L calculated using gallic acid. These sensors also exhibit a fast response time, an acceptable reproducibility and long-term stability. These results indicate that the proposed potentiometric sensor can be a promising and reliable tool for the rapid determination of total phenolic content in wine samples.

## 1. Introduction

Worldwide, wine consumption decreases the risk of cardiovascular disease and some cancers [1]. There is evidence that the presence of different phenolic substances, specifically those richly present in wine, might contribute to these biological effects on human health and disease prevention [2,3]. Aside from the well-recognized activity, phenolic compounds also contribute to sensorial characteristics of wines and the total phenolic content is also a worldwide standardized indicator to estimate the state of the quality of wine [4], therefore, rapid and accurate determination of total phenolic content in wine is of great importance for controlling sensory attributes and market value or quality.

Classical determination methods for total phenolic content in the laboratory rely on the Folin–Ciocalteu (FC) method, based on spectral detection. While this is a convenient and simple analytical technique for the total phenolic content in wine, it suffers from the limitations of not having an environmentally friendly reagent and a long processing time. Currently, a series of analytical methodologies based on infrared spectroscopy (IR), a chemiluminescence system and nuclear magnetic resonance (NMR) spectral have been developed for total phenolic content detection in a variety of samples [5,6,7,8]. Obviously, these tests cannot be performed easily worldwide due to their high cost. Mass spectrometric platforms targeting total phenols represent a burgeoning technology that facilitate the method development of qualitative and quantitative analysis with higher accuracy and a lower detection limit [9,10], however, these mass spectrometry-based platforms also have significant limitations, including a requirement for tedious sample pretreatment and sophisticated instruments, creating a high cost per sample. To compare, electrochemical sensors have been used as particularly attractive tools for total phenolic content analysis due to their high sensitivity, low manufacturing cost, fast response and ease of operation [11,12,13]. Electrochemical biosensors, based on the immobilization of laccase coupled with voltammetry, have been constructed successfully for rapid detection of total phenols [13,14], for example. Immobilization of enzymes, such as laccase, on electrodes requires complicated procedures, however, and is still a key challenge for operators. An alternative and highly successful approach, ion selective electrode-based potentiometry, has shown to be promising for trace-level measurements in food samples. A potentiometric methodology was fabricated for the determination of mono-phenols based on molecularly imprinted nanobeads as ionophores [15,16]. Unfortunately, the developed potentiometric strategies were not suitable for the determination of total phenol content. Recently, a label-free potentiometric biosensor based on solid-contact was fabricated for the determination of total phenols in honey and propolis, and the transducer-containing two layers was manufactured using the covalent bond method. Obviously, this platform also has significant limitations, including a requirement for tedious and complicated procedures and a high manufacturing cost [11]. Additionally, an ion-selective electrode was demonstrated for the assessment of the total content of polyphenolic antioxidants based on the use of CuII-neocuproin/2,6-dichlorophenolindo-phenolate [17], however, this method, with high detection limits of 6.3 to 9.2 g/mL, is not suitable for application in samples with a lower total phenol content.

Recently, a promising potentiometric detection approach based on ion fluxes across ion-selective electrode membranes has been found useful analytically for measuring some organic analytes which can decrease the concentrations of the indicator ions released at the membrane boundary via redox, complexing or enzyme-catalyzed reactions [18,19,20]. Currently, the potential change of the electrode is related to the concentration of the measured substance. Potentiometric analytical methods based on a permanganate release system, for example, have been developed for potentiometric detection of reductants such as dopamine and ascorbate [21,22]. Nevertheless, intense research efforts still focus on their new applications and, herein, ions for the evaluation of the total phenolic content in wine is the emphasis. Analysis conditions such as membrane composition, inner filling solution and pH are optimized. The results are compared with the data measured by the Folin–Ciocalteu (FC) method.

## 2. Results

### 2.1. Principle of Potential Response

Potassium permanganate, KMnO_4_, was found to be very lipophilic and showed a high anion response on the membrane electrode, based on the anion exchanger TDMAC. First of all, the potential response of MnO_4_^−^ on the fabricated electrode was investigated. Figure 1 illustrates, the proposed electrode shows a good Nernstian response of 58.34 mV/dec in the range from 10^−5^ to 10^−1^ M KMnO_4_. Gallic acid was chosen as a model of phenolic compounds of wine in this platform, since it was revealed to be one of the most abundant phenolic compounds in wine. Additionally, many methods developed for the determination of the total phenolic content were expressed as amounts of gallic acid in wine samples. Permanganate ion was used as the indicator ion for sensitive potentiometric detection. Illustrated in Scheme 1, inner permanganate ions of the indicator electrode accumulated at the sample-membrane phase boundary across a polymeric membrane using steady-state zero-current ion fluxes [18,23,24]. The presence of phenols in the sample solution, such as gallic acid, induced redox action and depletion of permanganate ions at the boundary of the electrode which led to a substantial charge change of the membrane-sample boundary and, therefore, a potential increase. The resultant potential changes were utilized for the determination of total phenolic content in wine.

### 2.2. Optimization of Analysis Conditions

#### 2.2.1. Membrane Composition

Lipophilic anion-exchangers such as tridodecylmethylammonium chloride (TDMAC) play a key role as added components of anion-membranes. They have been found to be responsible for extracting the analyte anions such as heparin polyion from the sample to the membrane [25]. Additionally, lipophilic mobile anion-exchanger sites of TDMAC play a key role as added components of anion-selective membranes. Their main function is to render the ion-selective membrane permselective, to optimize sensing selectivity and to reduce the bulk membrane impedance [26]. Membranes doped with different mass percentages of TDMAC were evaluated in that regard. Figure 2 shows, after gallic acid was added into the solution, the electrode exhibited larger potential changes when increasing the amount of TDMAC, up to 9.0%, which was probably due to an increase in the number of permanganate ions at the sample-membrane phase boundary. A further increase in the amount of TDMAC, however, would not improve the sensor’s sensitivity significantly. Thus, 9.0% was selected as the optimum.

#### 2.2.2. Inner Filling Solution

Recently, it has been realized fully that minor ionic fluxes in certain concentration ranges of the inner solution have an important role in determining the potentiometric response [27,28]. The effect of indicator permanganate ions on the detection sensitivity was investigated here and the result is illustrated in Figure 3. The concentration of inner permanganate ions was varied from 10^−3^ to 10^−1^ M, while that of gallic acid was fixed at 1.7 g/L. The electrode with a higher concentration of inner permanganate ions provided larger potential changes, as expected. Clearly, an electrode with inner permanganate ions at 10^−1^ M shows the largest potential changes of ~18 mV. It is well known that a higher concentration of inner filling solution induces the larger ion fluxes, which can facilitate the accumulation of permanganate ions at the boundary of the electrode membrane, thus causing higher potential responses. It might be concluded, therefore, that a concentration of inner permanganate ions at 10^−1^ M is sufficient for sensitive determination of the total phenolic content.

#### 2.2.3. pH

It is well known that the pH of a solution has a significant effect on the oxidation capacity of the oxidant and reducibility of the reductant, which will certainly induce potential changes of the electrode, based on redox reaction. Considering this fact, the influence of the pH of the solution on the potential response after redox reaction between permanganate ions and gallic acid was investigated. Considering the pH values of wine at the range of 3.0–4.0, the initial pH of the model wine was about 2.52 and adjusted to this range by use of 1.0 M NaOH. The results are presented in Figure 4. It can be seen clearly that the potential changes remained constant in the pH range of 3.0–4.0 when the concentration of gallic acid was varied from 0.17 to 1.7 g/L. These results indicate that the redox reaction between permanganate ions and gallic acid was not significantly influenced by pH and varied at the range of 3–4. A pH value of 3.6 was chosen for consistency of the proposed sensor.

### 2.3. Characteristics of the Proposed Potentiometric Sensor

The potential response curve of the proposed potentiometric sensor is shown in Figure 5. These data were obtained by adding standard solutions of 0.1 mL gallic acid to the test solution of 9.9 mL model wine solution. Illustrated in Figure 5, the potential response increased gradually with increasing concentrations of gallic acid, and detailed analysis of the experimental results indicated potential changes were found to be linear to the concentration of gallic acid in the range of 0.05 g/L to 3.0 g/L. This is mainly because the potential response is based on permanganate at a lower concentration and the response changes are proportional to the concentration of gallic acid, which is similar to other research work [22]. The total phenolic content was calculated by the regression equation y = (9.0748 ± 0.5457)x + (1.5958 ± 0.3225) with correlation coefficients R^2^ 0.9954. The detection limit is given by the equation C_L_ = 3s_bl_/S [29], where s_bl_ is the standard deviation of the blank measurements (s_bl_ = 0.02 mV) and S is the sensitivity of the calibration graph (S = 9.0748 mV/(g/L)). The detect limit of gallic acid was calculated to be 6.6 mg/L, which was satisfying for application of the proposed sensor to determine the total phenols in wine samples. Moreover, the proposed potentiometric sensor responded rapidly to the presence of gallic acid and achieved a steady potential response within less than 100 s.

A series of five potentiometric sensor measurements of 10 mM gallic acid were utilized for evaluating the detection precision. This series yielded reproducible potential changes with a relative standard deviation (RSD) of 6.4%, which confirmed an acceptable reproducibility of the fabrication sensor for analysis of real samples. The stability of the potentiometric system was also studied when these sensors were stored in a dry state at room temperature for a few days, and the results indicated that no significant potential change in sensitivity was observed for 20 days.

### 2.4. Analytical Application

The applicability of the proposed potentiometric sensor also was evaluated by determining the total phenolic content in six wine samples selected from the local market. A statistical comparison of the results from this sensor and the Folin–Ciocalteu method is presented in Figure 6. The concentration of the total phenolic compounds in the collected wine samples varied from 0.121 g/L~2.294 g/L. Additionally, the total phenolic content assessed by the proposed potentiometric sensor was higher compared to the data obtained by the Folin–Ciocalteu method which indicates that the proposed method overestimates the real phenolic content. This is mainly because the potentiometric sensor based on permanganate ion as an indicator also determines other reducing nonphenolic substances (e.g., sugars and ascorbic acid). Satisfactorily, the values of all test wine samples obtained from the two methods correlated highly and the Pearson correlation coefficient r was 0.8535. These results indicate that the proposed potentiometric sensor can be a promising and reliable tool for the rapid determination of total phenolic content in wine samples.

## 3. Materials and Methods

### 3.1. Reagents and Materials

High molecular weight poly(vinyl chloride) (PVC), 2-nitrophenyl octyl ether (o-NPOE), tridodecylmethylammonium chloride (TDMAC), tetrahydrofuran (THF) and gallic acid (GA) were purchased from Sigma–Aldrich (St. Louis, MO, USA). Potassium permanganate, tartaric acid, sodium tartrate, sodium chloride, lithium acetate and ethanol were obtained from Sinopharm Chemical Reagent Co., Ltd. (Shanghai, China). A stock solution of 0.1 M for gallic acid was prepared with the model wine solution (12% vol ethanol, 4 g/L tartaric acid, pH 3.6). A 0.1 M Potassium permanganate was prepared daily and stored in the dark. Other aqueous solutions were prepared by dissolving the appropriate salts in the freshly de-ionized water (18.2 MΩ cm specific resistance) obtained with a Pall Cascada laboratory water system.

### 3.2. Potentiometric Sensor Preparation

The ionophore-free membrane of ~210 μm thickness was prepared by dissolving 200 mg of 9 wt.% TDMAC, 31 wt.% PVC and 60 wt.% *o*-NPOE in 2.0 mL of THF. The membrane cocktail was degassed by sonication for 10 min before use and then poured into a glass ring (26 mm i.d.) fixed on a glass plate. Subsequently, these were completely air-dried. Then, 6-mm-diameter membrane disks were cut from the membrane and glued to a plasticized PVC tube with THF/PVC slurry. 10^−1^ M KMnO_4_ was used as the inner filling solution for each electrode. Prior to measurements, all the electrodes were conditioned overnight in 10 mM of NaCl.

### 3.3. Potentiometric Measurements

All electromotive force measurements (EMF) were carried out at 25 °C using a CHI 760D electrochemical workstation (Shanghai, China) in the galvanic cell: SCE/1 M LiOAC/sample solution/ISE membrane/inner filling solution/Ag, AgCl. The open circuit potential (OCP) of the MnO_4_^−^ based potentiometric sensor recorded the model wine solution used as the baseline, then, 0.1 mL of gallic acid at different concentrations was added into 9.9 mL of the model wine solution and the potential response was recorded. The potential change at 100 s was used for quantification of total phenolic contents.

### 3.4. Application of Proposed Potentiometric Sensor

The wine samples were collected from local markets and their total phenolic contents were determined by the present potentiometric sensor in the same manner that 0.1 mL of wine was added into 9.9 mL of the model wine solution. Total phenolic content concentrations were calibrated from the standard curve between gallic acid concentration and MnO_4_^−^ potential response.

## 4. Conclusions

A simple and robust potentiometric approach for determining the total phenolic content has been successfully proposed. The quantitative analysis method is based on the potential changes induced by redox action between permanganate ion fluxes across the polymeric membrane and phenols such as gallic acid in the sample solution. The proposed electrodes demonstrate to be linear to the concentration of gallic acid in the range of 0.05 g/L to 3.0 g/L and the detection limit is 6.6 mg/L. Additionally, these also exhibit a fast response time, an acceptable reproducibility and long-term stability. Note that, although the total phenolic content assessed by the proposed potentiometric sensor was higher compared to the data obtained by the Folin–Ciocalteu method, the values from the two methods correlate highly and the Pearson correlation coefficient r is 0.8535. Therefore, this new approach might pave the way to detect and quantify total phenolic content in other food analysis applications.

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
