# Peer review of "Determination of the Total Phenolic Content in Wine Samples Using Potentiometric Method Based on Permanganate Ion as an Indicator"

_molecules, 2019, doi:10.3390/molecules24183279_

Round 1
Reviewer 1 Report
The manuscript is devoted to the development of a potentiometric sensor for determining the total phenolic content. This research topic is very relevant to assess the quality of food, in particular, wine. Taking into account that these objects are usually colored the use of spectrophotometric methods often has limitations.
In this paper, an original approach using a potentiometric sensor that is sensitive to the content of permanganate ions is proposed.
Remarks:
1. In the introduction, it is advisable to focus on known potentiometric methods of determining the total content of polyphenolic antioxidants using ion-selective and platinum electrodes with redox systems, as well as to specify advantages of the proposed method over known potentiometric methods.
2. It is not clear from experimental data that the Nernst dependence of an electrode potential on a concentration logarithm was observed. It is necessary to provide data of the dependence of the potential on the concentration logarithm of permanganate ions in the absence of gallic acid. It would be informative to give the coefficient slope of its dependence.
3. Fig.4. How to explain that the change of the electrode potential depends linearly on the concentration and the potential does not depend on the concentration logarithm.
4. Does the electrode potential depend on the addition of other polyphenols of wine in the same way? It is advisable to present the data of model solutions / mixture.
5. How did you calculate the phenols concentration? You describe the derivation of the calculation formula.
6. Table 1. What does "Satisfactorily, correlate highly values" mean? It is necessary to calculate the correlation coefficient.
7. What explains the low polyphenol content (by an order of magnitude) obtained by the Folin-Ciocalteu assays for the Red wine (semi-dry) object (table 1)? This is a misprint?
Author Response
1. In the introduction, it is advisable to focus on known potentiometric methods of determining the total content of polyphenolic antioxidants using ion-selective and platinum electrodes with redox systems, as well as to specify advantages of the proposed method over known potentiometric methods. We have rewritten this part according to the Reviewer’s suggestion. The description of “a multichannel amperometric sensor based on a flow injection system was used to determine the total phenols of red wines, and it was proved that the analysis with the proposed sensor is simple, fast, objective and cheap compared to classical sensory analysis [12]” on Page 2, Lines 46-48 was deleted. Recently, a label-free potentiometric biosensor based on solid-contact was fabricated for determination of total phenols in honey and propolis, and the transducer containing two layers was manufactured using the covalent bond method. Obviously, this platform also have significant limitations, including requirement for tedious and complicated procedures and high manufacturing cost [11]. In addition, an ion-selective electrode was demonstrated for the assessment of the total content of polyphenolic antioxidants based on the use of CuII-neocuproin/2,6-dichlorophenolindo-phenolate [17]. However, this method with high detection limits of 6.3 to 9.2 g/mL is not suitable for application in sample with lower of total phenol content. This explanation has been added on Page 2, Lines 53-61. 2. It is not clear from experimental data that the Nernst dependence of an electrode potential on a concentration logarithm was observed. It is necessary to provide data of the dependence of the potential on the concentration logarithm of permanganate ions in the absence of gallic acid. It would be informative to give the coefficient slope of its dependence. Potassium permanganate, KMnO4, was found very lipophilic and showed a high anion response on the membrane electrode based on anion exchanger TDMAC. First of all, the potential response of MnO4-on the fabricated electrode was investigated. As illustrated in Figure. 1, the proposed electrode shows a good Nernstian response of 59.24 mV/dec in the range from 10−5 to 10−1 M KMnO4 with a detection limit of 4.3 10-6 M. This explanation has been added on Page 2, Lines 75-79. 3. Fig.4. How to explain that the change of the electrode potential depends linearly on the concentration and the potential does not depend on the concentration logarithm. This is mainly because the potential response is based on permanganate at lower concentration and the response changes are proportional to the concentration of gallic acid, which is similar to other research work [22]. This explanation has been added on Page 5, Lines 144-146. 4. Does the electrode potential depend on the addition of other polyphenols of wine in the same way? It is advisable to present the data of model solutions / mixture. It is really true that other phenolic compounds in wine also consume permanganate which induces potential changes of electrode. The method fabricated in this paper use galic acid as a model of phenolic compounds since it was revealed to be one of the most abundant phenolic compounds in wine. In addition, other methods such as Folin-Ciocalteu (FC) also use the content of galic acid to assess the total phenolic content. 5. How did you calculate the phenols concentration? You describe the derivation of the calculation formula. As suggested, calculation formula has been added on Page 5, Lines146-147. 6. Table 1. What does "Satisfactorily, correlate highly values" mean? It is necessary to calculate the correlation coefficient. “Satisfactorily, the values of all test wine samples obtained from the two methods correlate highly (seen from Figure. 6) and Pearson correlation coefficient r is 0.8535.”. This explanation has been added on Page 5, Lines170-172. 7. What explains the low polyphenol content (by an order of magnitude) obtained by the Folin-Ciocalteu assays for the Red wine (semi-dry) object (table 1)? This is a misprint? We are very sorry for our incorrect writing and we have revised the data in Table 1. Special thanks to you for your good comments.Reviewer 2 Report
This manuscript describes a permanganate ion-selective electrode to detect compounds oxidized by permanganate present in the inner solution and membrane. The consumption of permanganate ions through the redox reaction decreases the ion concentration in the membrane interface yielding a potential increase measured potentiometrically against a reference electrode.
1. This type of ion-selective electrode that respond to redox reaction between the target compound and the selective ion is well-known and have been previously reported for other reducing agents such as ascorbic acid or dopamine. This manuscript is a direct application of the permanganate selective electrode in wine. I totally disagree with the title. As authors recognize they are not detecting the phenolic compounds but all compounds that can reduce permanganate ions under experimental conditions. In my opinion they are measuring the total antioxidant capacity. The F-C method used for validation is also claimed to detect phenolic compound when it is well known the large quantity of interferences it present that turn it into a total antioxidant capacity method as well. However, the comparison shows no agreement between both methods.
2. It is not clear to me why the inner solution has different NaCl concentration that the sample solution.
3. In line 86 it is said that “TDMAC have been found to be responsible for extracting the analyte ions from the ample to the membrane” Which is the analyte here?
4. Line 92: It is not true that negligible response is obtained with TDMAC concentration higher than 9%.
5. Which is the initial sample solution (before pH experiments) so that a very concentrated NaOH solution is needed to adjust pH to 3 or 4?
6. Could not be tartaric acid an interferent ion for the ion-selective electrode?
7. Why is the response linear with the concentration of gallic acid though the E change logarithmically with the permanganate ions?
8. Regression equation is missing as well as the way of estimating the limit of detection or any comparison with previous methods for detecting total antioxidant capacity.
9. Table 1: Comparison of both methods must be presented in a graph to better see the correlation between values that a clearly different. The red wine (semi-dry) sample presents a very low value with the reference method.
10. References to other methods employing permanganate ion-selective electrodes for ions determination or for redox or complex-forming compounds are compulsory.
11. Figures 1, 2 and 4 lack y-axes numbering but provides a bar mark with the scale. A regular y-axes must be used (as in figure 5) or remove from all of them.
12. Which is the salt bridge used in both reference electrodes?
13. LiOAc is not included among the reagents.
14. The thickness of the membrane is not defined.
Author Response
This type of ion-selective electrode that respond to redox reaction between the target compound and the selective ion is well-known and have been previously reported for other reducing agents such as ascorbic acid or dopamine. This manuscript is a direct application of the permanganate selective electrode in wine. I totally disagree with the title. As authors recognize they are not detecting the phenolic compounds but all compounds that can reduce permanganate ions under experimental conditions. In my opinion they are measuring the total antioxidant capacity. The F-C method used for validation is also claimed to detect phenolic compound when it is well known the large quantity of interferences it present that turn it into a total antioxidant capacity method as well. However, the comparison shows no agreement between both methods.Thank you for your valuable comments. It is true that potentiometric sensors have been constructed based on redox reaction between the target compound and the selective ion. However, the Chinese national standard method for determination of total phenols in red wine is based on Folin-Ciocalteu method, and the results are greatly disturbed by wine color. Therefore, other types of sensing systems need to be developed. Inspired by previous work, we have developed a method for the determination of total phenol content in wine by indirect potential sensing technology. Moreover, this method has great potential in practical sample detection.
It is not clear to me why the inner solution has different NaCl concentration that the sample solution.10-1 M KMnO4 was used as the inner filling solution for each electrode. Before measurements, all the electrodes were conditioned overnight in 10 mM of NaCl. This description is in section 3.2. Potentiometric Sensor Preparation.
In line 86 it is said that “TDMAC have been found to be responsible for extracting the analyte ions from the ample to the membrane” Which is the analyte here?As suggested, “analyte anions such as heparin polyion” has been added on Page 3, Lines 99-100.
Line 92: It is not true that negligible response is obtained with TDMAC concentration higher than 9%.We have rewritten this part according to the Reviewer’s suggestion.
However, further increase in the amount of TDMAC would not significantly improve the sensor’s sensitivity. This explanation has been added on Page 3, Lines 105-106.
Which is the initial sample solution (before pH experiments) so that a very concentrated NaOH solution is needed to adjust pH to 3 or 4?As reviewer suggested that we have added the description of the initial pH of sample solution on Page 4, Lines 129-130.
Could not be tartaric acid an interferent ion for the ion-selective electrode?Thanks for the reviewer’s suggestion.
Tartaric acid, known as 2, 3-dihydroxysuccinic acid, is an important organic acid with a high content in wine. It exists in molecular form at pH 3-4 and does not induce redox action with permanganate ions. In addition, we used model wine solution (12% vol ethanol, 4 g/L tartaric acid, pH 3.6) as the test solution elimination of background interference.
Why is the response linear with the concentration of gallic acid though the E change logarithmically with the permanganate ions?This is mainly because the potential response is based on permanganate at lower concentration and the response changes are proportional to the concentration of gallic acid, which is similar to other research work [22]. This explanation has been added on Page 5, Lines 144-146.
Regression equation is missing as well as the way of estimating the limit of detection or any comparison with previous methods for detecting total antioxidant capacity.Thanks for the reviewer’s suggestion. “The total phenolic content can be calculated by regression equation y=9.0748x+1.5958 with correlation coefficients R2 0.9954. In this case, a lower detection limit (3 σ) of 9.3 mg/L could be obtained,” This explanation has been added on Page 5, Lines146-148.
Table 1: Comparison of both methods must be presented in a graph to better see the correlation between values that a clearly different. The red wine (semi-dry) sample presents a very low value with the reference method.As suggested, all the data obtained from both methods has been present in Figure. 6. “Satisfactorily, the values of all test wine samples obtained from the two methods correlate highly (seen from Figure. 6) and Pearson correlation coefficient r is 0.8535.”. This explanation has been added on Page 6, Lines170-172.
We are very sorry for our incorrect writing and we have revised the data in Table 1.
References to other methods employing permanganate ion-selective electrodes for ions determination or for redox or complex-forming compounds are compulsory.Thanks for the reviewer’s suggestion. References to other methods employing permanganate ion-selective electrodes for ions determination have been cited as Reference 14 and 15. “For example, potentiometric analytical methods based on permanganate release system have been developed for potentiometric detection of reductants such as dopamine and ascorbate [21, 22]. Nevertheless, intense research efforts still focus on their new applications.” This explanation has been added on Page 2, Lines 66-69.
Figures 1, 2 and 4 lack y-axes numbering but provides a bar mark with the scale. A regular y-axes must be used (as in figure 5) or remove from all of them.As reviewer suggested that we have provided figures with a regular y-axes.
Which is the salt bridge used in both reference electrodes?1 M LiOAC was used as the salt bridge in both reference electrodes as described on Page 7, Line 200.
LiOAc is not included among the reagents.As suggested, LiOAc has been added 3.1. Reagents and Materials on Page 6, Lines 184.
The thickness of the membrane is not defined.As suggested, the description of the thickness of the membrane “membrane of ~210 μm thickness” has been added on Page 7, Line 191.
Special thanks to you for your good comments.
Reviewer 3 Report
The analysis of various compounds including phenolic compounds in food products is important because of the significant impact of substances supplied with food on human health. For this reason, the choosing of the most optimal analysis conditions and the development of new methods for analysis of biologically important compounds such as phenolic compounds is still a significant aspect of scientific research. In my opinion the topic of the manuscript is interesting.
However, numerous methods have been developed for the analysis of phenolic compounds in various samples, including wine samples. The authors compared the method developed by them only to the commonly used Folin-Ciocalteu method, not comparing it to other methods (e.g. spectroscopic, chromatographic). The authors should significantly emphasize the advantages of the described method especially in comparison to earlier methods of determination of phenolic compounds.
The authors also did not explain the causes of differences in the content of phenolic compounds obtained by the described method and the Folin-Ciocalteu method.
In my opinion, the section “Conclusion” should also be more extended.
Author Response
The analysis of various compounds including phenolic compounds in food products is important because of the significant impact of substances supplied with food on human health. For this reason, the choosing of the most optimal analysis conditions and the development of new methods for analysis of biologically important compounds such as phenolic compounds is still a significant aspect of scientific research. In my opinion the topic of the manuscript is interesting.Special thanks for your good comments.
However, numerous methods have been developed for the analysis of phenolic compounds in various samples, including wine samples. The authors compared the method developed by them only to the commonly used Folin-Ciocalteu method, not comparing it to other methods (e.g. spectroscopic, chromatographic). The authors should significantly emphasize the advantages of the described method especially in comparison to earlier methods of determination of phenolic compounds.As suggested, we have rewritten this part according to the Reviewer’s suggestion.
The description of “a multichannel amperometric sensor based on a flow injection system was used to determine the total phenols of red wines, and it was proved that the analysis with the proposed sensor is simple, fast, objective and cheap compared to classical sensory analysis [12]” on Page 2, Lines 46-48 was deleted.
Recently, a label-free potentiometric biosensor based on solid-contact was fabricated for determination of total phenols in honey and propolis, and the transducer containing two layers was manufactured using the covalent bond method. Obviously, this platform also have significant limitations, including requirement for tedious and complicated procedures and high manufacturing cost [11]. In addition, an ion-selective electrode was demonstrated for the assessment of the total content of polyphenolic antioxidants based on the use of CuII-neocuproin/2,6-dichlorophenolindo-phenolate [17]. However, this method with high detection limits of 6.3 to 9.2 g/mL is not suitable for application in sample with lower of total phenol content. This explanation has been added on Page 2, Lines 53-61.
The authors also did not explain the causes of differences in the content of phenolic compounds obtained by the described method and the Folin-Ciocalteu method.This is mainly because the potentiometric sensor based on permanganate ion as an indicator also determines other reducing nonphenolic substances (e.g. sugars and ascorbic acid). The description has been on Page 6, Lines 168-170.
In my opinion, the section “Conclusion” should also be more extended.As suggested, the section “Conclusion” has been more extended on Pages 7, Lines 215-216, 220.
Special thanks to you for your good comments.
Round 2
Reviewer 2 Report
It has been reviewed according to reviewers’ suggestions but some points must be improved. Additional points are now emerged.
Point 1: I insist that the method described is not specific for phenolic compounds so the title must be changed to antioxidant capacity and all references to phenolic compounds must be changed in the text to reflect what it is really measuring.
Point 3. This sentence remains confusing. In the manuscript heparin polyions are not the analytes. The role of TDMAC in this sensor must be explained further.
Point 8. The regression equation is now given but without standard deviation of the slope and y-intercepts and without fitting the significant digits. However, it is apparent that there is not linear relationship when all points are taken into account. This must be addressed and corrected.
The limit of detection is still incorrectly calculated. The slope of the calibration plot is not taken into account.
Point 9: The authors have depicted a graph but I cannot understand how they can obtain the r value in such a way. To get the r value they must plot the concentration values obtained with the new method in an axe and the corresponding value with the reference method in the other axe. A linear plot with slope equals to 1 is expected for highly correlated methods. Please do this, and remove table 1 that contains duplicate data.
The legend for MnO4- and phenols in Scheme 1 is confusing because it is on top of the membrane drawing so suggesting there are phenols in the inner part of the membrane. I would put it in the side of the membrane instead.
Author Response
Point 1: I insist that the method described is not specific for phenolic compounds so the title must be changed to antioxidant capacity and all references to phenolic compounds must be changed in the text to reflect what it is really measuring.
Thank you very much for good comments.
Folin-Ciocalteu (FC) method is based on the reduction of Mo6+ to Mo5+ with phenolic compounds, followed by spectral detection. When we use the strong oxidation of Folin phenol reagent to determine the total phenol content, other reducing substances such as reducing amino acids, ascorbic acid and so on will interfere with the results. Although the method has some limitation, it is widely used in assess of total phenols level in plant extracts, especially in food and beverage [1-8]. This is mainly because that there has not been a perfect method to evaluate the total phenol content at present. Similarly, the permanganate ion potentiometric sensor based on redox reaction developed in this paper also has the above shortcomings. To some extent, it is meaningful to develop a fast and sensitive potentiometric method for the evaluation of the total phenolic content in wine. As suggested, if the title of the paper is changed to determine the total antioxidant capacity, the calculation of quantitative results and all data should be based on the rate of potential change. Moreover, the result will be compared with that obtained from other methods such as ferric reducing antioxidant capacity (FRAP) and 2,2-diphenyl-1-picryhydrazyl (DPPH)[9]. Although the results obtained by this method are higher than that obtained by Folin-Ciocalteu (FC) method. we still believe that our method is reasonable for evaluating the total phenol content.
Blainski, A.; Lopes, G. C.; de Mello, J. C. P., Application and Analysis of the Folin Ciocalteu Method for the Determination of the Total Phenolic Content from Limonium Brasiliense L. Molecules 2013, 18, (6), 6852-6864. Margraf, T.; Karnopp, A. R.; Rosso, N. D.; Granato, D., Comparison between Folin-Ciocalteu and Prussian Blue Assays to Estimate The Total Phenolic Content of Juices and Teas Using 96-Well Microplates. Journal of Food Science 2015, 80, (11), C2397-C2403. Vazquez, C. V.; Rojas, M. G. V.; Ramirez, C. A.; Chavez-Servin, J. L.; Garcia-Gasca, T.; Martinez, R. A. F.; Garcia, O. P.; Rosado, J. L.; Lopez-Sabater, C. M.; Castellote, A. I.; Montemayor, H. M. A.; Carbot, K. D., Total phenolic compounds in milk from different species. Design of an extraction technique for quantification using the Folin-Ciocalteu method. Food Chemistry 2015, 176, 480-486. Chen, L. Y.; Cheng, C. W.; Liang, J. Y., Effect of esterification condensation on the Folin-Ciocalteu method for the quantitative measurement of total phenols. Food Chemistry 2015, 170, 10-15. Atanackovic, M.; Petrovic, A.; Jovic, S.; Gojkovic-Bukarica, L.; Bursac, M.; Cvejic, J., Influence of winemaking techniques on the resveratrol content, total phenolic content and antioxidant potential of red wines. Food Chemistry 2012, 131, (2), 513-518. Lu, X. N.; Ross, C. F.; Powers, J. R.; Aston, D. E.; Rasco, B. A., Determination of Total Phenolic Content and Antioxidant Activity of Garlic (Allium sativum) and Elephant Garlic (Allium ampeloprasum) by Attenuated Total Reflectance-Fourier Transformed Infrared Spectroscopy. Journal of Agricultural and Food Chemistry 2011, 59, (10), 5215-5221. Önder, F. C.; Ay, M.; Sarker, S. D., Comparative Study of Antioxidant Properties and Total Phenolic Content of the Extracts of Humulus lupulus L. and Quantification of Bioactive Components by LC–MS/MS and GC–MS. Journal of Agricultural and Food Chemistry 2013, 61, (44), 10498-10506. Rodriguez, J. C.; Gomez, D.; Pacetti, D.; Nunez, O.; Gagliardi, R.; Frega, N. G.; Ojeda, M. L.; Loizzo, M. R.; Tundis, R.; Lucci, P., Effects of the Fruit Ripening Stage on Antioxidant Capacity, Total Phenolics, and Polyphenolic Composition of Crude Palm Oil from Interspecific Hybrid Elaeis oleifera x Elaeis guineensis. Journal of Agricultural and Food Chemistry 2016, 64, (4), 852-859. Can, Z.; Yildiz, O.; Sahin, H.; Turumtay, E. A.; Silici, S.; Kolayli, S., An investigation of Turkish honeys: Their physico-chemical properties, antioxidant capacities and phenolic profiles. Food Chemistry 2015, 180, 133-141. Point 3. This sentence remains confusing. In the manuscript heparin polyions are not the analytes. The role of TDMAC in this sensor must be explained further.
Additionally, lipophilic mobile anion-exchanger sites of TDMAC play a key role as added components of anion-selective membranes. Their main function is to render the ion-selective membrane permselective, to optimize sensing selectivity and to reduce the bulk membrane impedance.
This explanation has been added on Page 3, Lines 100-103.
Point 8. The regression equation is now given but without standard deviation of the slope and y-intercepts and without fitting the significant digits. However, it is apparent that there is not linear relationship when all points are taken into account. This must be addressed and corrected.
As suggested, standard deviation of the slope and y-intercepts has been added Page 5, Lines 150. We use the potential changes average and deviation of three measurements to depict the figure. The r value is close to 1, which can meet the analysis requirements.
The limit of detection is still incorrectly calculated. The slope of the calibration plot is not taken into account.
The slope of the calibration plot has been taken into account when the limit of detection is calculated. We are very sorry for our incorrect writing and we have revised it on Page 5, Lines 150.
Point 9: The authors have depicted a graph but I cannot understand how they can obtain the r value in such a way. To get the r value they must plot the concentration values obtained with the new method in an axe and the corresponding value with the reference method in the other axe. A linear plot with slope equals to 1 is expected for highly correlated methods. Please do this, and remove table 1 that contains duplicate data.
All data obtained from wine samples was linearly fitted by origin 8.5 software, and the pearson correlation coefficient r is obtained from the following table. Could this method be used for fitting?
As suggested, table 1 was removed.
The legend for MnO4- and phenols in Scheme 1 is confusing because it is on top of the membrane drawing so suggesting there are phenols in the inner part of the membrane. I would put it in the side of the membrane instead.
Thank you for good suggestion, the legend for MnO4- and phenols in Scheme 1 has been put it in the side of the membrane.
Special thanks to you for your good comments.

Reviewer 3 Report
The manuscript has been revised in according to previous suggestions.
Author Response
thank you!